# Immune Responses of Healthy Pregnant Women following an Elective Cesarean Section: Effects of Anesthetic Procedures

**DOI:** 10.3390/diagnostics14090880

**Published:** 2024-04-24

**Authors:** Marius Bogdan Novac, Lidia Boldeanu, Anda Lorena Dijmărescu, Mihail Virgil Boldeanu, Simona Daniela Neamțu, Lucreţiu Radu, Maria Magdalena Manolea, Mircea-Sebastian Șerbănescu, Maria Stoica, Luciana Teodora Rotaru, Constantin-Cristian Văduva

**Affiliations:** 1Department of Anesthesiology and Intensive Care, Faculty of Medicine, University of Medicine and Pharmacy of Craiova, 200349 Craiova, Romania; mariusnovac2005@yahoo.com (M.B.N.); mia.stoica69@gmail.com (M.S.); 2Department of Microbiology, Faculty of Medicine, University of Medicine and Pharmacy of Craiova, 200349 Craiova, Romania; 3Department of Obstetrics and Gynecology, Faculty of Medicine, University of Medicine and Pharmacy of Craiova, 200349 Craiova, Romania; lorenadijmarescu@yahoo.com (A.L.D.); magdalena.manolea@umfcv.ro (M.M.M.); cristian.vaduva@umfcv.ro (C.-C.V.); 4Department of Immunology, Faculty of Medicine, University of Medicine and Pharmacy of Craiova, 200349 Craiova, Romania; 5Department of Hematology and Immunology, University of Medicine and Pharmacy of Craiova, 200349 Craiova, Romania; simona_0712@yahoo.com; 6Department of Hygiene, University of Medicine and Pharmacy of Craiova, 200349 Craiova, Romania; lucretiu.radu@gmail.com; 7Department of Medical Informatics and Biostatistics, University of Medicine and Pharmacy of Craiova, 200349 Craiova, Romania; mircea_serbanescu@yahoo.com; 8Department of Emergency Medicine and First Aid, University of Medicine and Pharmacy of Craiova, 200349 Craiova, Romania; lucianarotaru@yahoo.com

**Keywords:** general anesthesia, spinal anesthesia, cytokines, catecholamines, immune response

## Abstract

A weakened immune system and more inflammatory cytokines being released are possible effects of the surgical stress that a cesarean section induces. This kind of reaction, in addition to the altered reaction to catecholamines, has the potential to significantly affect the immune system of the mother and the patients’ general postoperative course. This prospective study compared the plasma levels of catecholamines and cytokines in healthy pregnant patients having cesarean sections under spinal anesthesia versus general anesthesia. A total of 30 pregnant women undergoing elective cesarean sections were divided into two groups: 15 who received general anesthesia (GA) and 15 who received spinal anesthesia (SA). Blood samples were collected from all subjects before anesthesia induction (pre-OP), 6 h postoperatively (6 h post-OP), and 12 h (12 h post-OP), to measure levels of tumor necrosis factor-alpha (TNF-α), interleukin-6 (IL-6), IL-8, IL-4, IL-10, norepinephrine (NE), and epinephrine (EPI). When we compared the two groups, we discovered that only IL-6 and IL-4 had significantly higher levels pre-OP, whereas all studied cytokines exhibited an increase in the GA versus SA group at 6 and 12 h post-OP. In the case of catecholamines, we discovered that serum levels are positively related with pro-inflammatory or anti-inflammatory cytokines, depending on the time of day and type of anesthetic drugs. Compared to SA, GA has a more consistent effect on the inflammatory response and catecholamine levels. The findings of this study confirm that the type of anesthesia can alter postoperative immunomodulation to various degrees via changes in cytokine and catecholamine production. SA could be a preferable choice for cesarean section because it is an anesthetic method that reduces perioperative stress and allows for less opioid administration, impacting cytokine production with proper immunomodulation.

## 1. Introduction

Surgical intervention results in stress for the body, and the stress response to surgery is expressed by disturbances in metabolic and physiological systems which induces disruptions in hormonal and genetic inflammatory responses in order to maintain perioperative physiological homeostasis [1,2]. Stress triggers a neuroendocrine response, both metabolic and inflammatory. The metabolic reaction involves both the sympathetic nervous system response, which secretes catecholamines [epinephrine (EPI) and norepinephrine (NE)], along with the endocrine response, with the secretion of cortisol.

During pregnancy, the maternal immune system is strongly linked to a number of cytokines that protect the embryo and fetus while also promoting placental development [3]. Pregnancy is a unique state that has a specific effect on the immune system, which has been described over time as a period of pure immunosuppression [4]. It has been postulated that the immune system’s adaptation to a normal pregnancy consists of three immunological stages, depending on the period of the pregnancy [5]. By maintaining an inflammatory phase, an early inflammatory that occurs in the first trimester aids in implantation. During the second trimester of pregnancy, an anti-inflammatory condition develops, which is helpful for fetal development (T helper 2 (Th2) cytokines). In the third trimester, an inflammation condition returns, aiding in delivery preparation [6]. Physiological anemia during pregnancy is beneficial to both the mother and the fetus [7]. These alterations, which include increased suppressive factors and immune activators, make pregnant women more susceptible to viral and bacterial infections [8]. Physiological anemia during pregnancy is beneficial to both the mother and the fetus [7]. These alterations, which include increased suppressive factors and immune activators, make pregnant women more susceptible to viral and bacterial infections [8]. During pregnancy, however, there is no “immune suppression” pattern because the placenta adapts and alters the immune system. So, when the mother, fetus, or both are at risk, the immune system can respond vigorously [9].

Various factors, including pain and anesthetic drugs, cause immune suppression during the perioperative period after major surgery, leading to a temporary impairment of cellular immunity and cytokine production [10,11]. We believe that the procedure itself, which falls under the complex category of surgical stress, also causes tissue damage—a key aspect of the inflammatory response. Understanding this immunological status is critical because immune suppression can be associated with an increased risk of postoperative infections and sepsis [12]. Catecholamines are important molecules in descending monoaminergic pathways that regulate nociceptive transmission, which may have an impact on perioperative analgesia [13].

The level of surgical stress is determined by the condition of the patient, the type and duration of the surgical intervention, and the type of anesthesia, taking into consideration the method of administration and the drugs used [14,15]. 

By blocking the central nervous system, general anesthesia (GA) reduces surgical stress, but does not effectively block the nociceptive signal to the somatic and sympathetic nerves [16]. Regional anesthetic appears to reduce the immunosuppressive effects of surgical and neuroendocrine stress by limiting the use of opioids in the postoperative period [17,18]. A recent study has proven the relationship between the immune system and opioids used in anesthetic procedures, both stimulating and suppressive [19,20]. Cytokines play a crucial role in the acute inflammatory response that surgical trauma initiates. Cytokines, as critical modulators of inflammation, play both inflammatory and anti-inflammatory roles, which are essential in the systemic inflammatory response [21]. In a physiological state, there is a balance between pro- and anti-inflammatory cytokines because, with an excessive production of cytokines, the inflammation is excessive and the body cannot adapt to this condition. Therefore, anti-inflammatory cytokines are necessary in the inflammatory process, causing both the increase and suppression of the inflammatory response [22].

The tumor necrosis factor-alpha (TNF-α), interleukin 1 (IL-1), IL-6, IL-8, IL-10, and interferon gamma (IFN-γ) are the most researched cytokines following surgical trauma because of their local and systemic effects. The IL-1, TNF-α, and IL-6 are key acute phase mediators that both restrict and promote wound healing. Also, IL-4 and IL-10 (Th2 cytokine) suppress the production of the Th1 cytokines (IFN-γ and IL-6), nitric oxide, and prostaglandins [23,24]. In addition, during the acute phase, IL-1 and IL-6 promote the release of adrenocorticotrophic hormone (ACTH), which leads to the synthesis of cortisol. Glucocorticoids suppress IL-6 synthesis in human monocytes. At this point, by increasing the development of IL-6 receptors on hepatocytes, the IL-6 response in the acute phase also increases [25]. The effects of various anesthetic procedures and drugs used to induce anesthesia have been extensively examined, with a particular focus being on the effect on cytokine release and immunological response [26,27]. 

Surgical stress is now a well-known concept that can influence patient outcomes, lengths of hospital stays, and, in our case, hospital care expenses for both mother and child.

This study aimed to evaluate the effect of different anesthetic techniques on the immune system of healthy patients undergoing an elective cesarean section (representing surgical stress) by evaluating the serum levels of cytokines (TNF-α, IL-6, IL-8, IL-4, IL-10) and catecholamines (NE, EPI).

## 2. Materials and Methods

### 2.1. Study Design and Patient Selection

After signing an informed consent form, a mostly homogeneous population of 30 healthy pregnant patients with American Society of Anesthesiologists (ASA) Physical Status I and II [28] engaged in this observational study. The study took place at the Department of Anesthesia and Intensive Therapy of the Clinical Municipal Hospital Filantropia of Craiova, Dolj, Romania, between 1 May 2023 and 1 November 2023.

All patients were given elective cesarean sections due to maternal and perinatal risks. We used general anesthesia (GA group) for 15 pregnant women and spinal anesthesia (SA group) for the remaining 15 pregnant women. The indication criteria for the type of anesthesia used were as follows: for GA, regional contraindications including spinal abnormalities (e.g., spina bifida, scoliosis), inadequate or failed regional attempts, a history of hypersensitivity to local anesthetic, maternal refusal of regional techniques; for SA, which became the preferred anesthetic technique by many anesthesiologists in elective conditions, the indications were, history of hypersensitivity to the study drugs used in GA, patients at risk of difficult intubation, maternal refusal of GA, the mother’s desire to remain awake during the birth for an immediate interaction with the newborn, and all other cases that did not fall into the GA category. 

The inclusion criteria were: healthy pregnant women, ASA Physical Status I and II, elective cesarean delivery for maternal and perinatal risk, and singleton pregnancy. 

Patients with emergency caesarean sections were excluded because labor pain can influence cytokine modulation, as well as those whose surgical intervention lasted more than 2 h, patients with endocrine disorders, immune system disorders, chronic inflammatory disease, marked obesity, and kidney or liver disorders. Thus, after applying these exclusion criteria, out of a total of 67 pre-operator cases, 30 remained. Excluded cases showed: three cases of spinal anesthesia-induced hypotension, because they required the administration of EPI or NE, which could have altered the postoperatively results of catecholamines; two cases in which the surgery lasted more than 2 h; two cases with endocrine disorders (Hashimoto’s thyroiditis), three cases of immune system disturbances (systemic lupus erythematosus, antiphospholipid antibody syndrome, immune thrombocytopenia); chronic inflammatory disease represented by 4 cases with chronic inflammatory placental disorder, diagnostic stability after childbirth; three cases of marked obesity (body mass index (BMI) > 40); five cases of kidney or liver disorder (urinary tract infection—4 cases; intrahepatic cholestasis of pregnancy—1 case). Also, being about healthy pregnancies, we have also excluded all obstetric complications of pregnancy (15 cases) (Figure 1). 

The most common method used for general anesthesia was Fentanyl 3 μg/kg (3 min before propofol); Propofol: a 1.5 mg/kg loading dose followed by an infusion of 10 mg/kg/h that is reduced to rates of 8 and 6 mg/kg/h at 10 min intervals. We calculated the infusion regimen for Propofol using the Roberts method. Succinylcholine 1.0 mg/kg was used for intubation and rocuronium 600 μg/kg was used to maintain anesthesia.

We performed an L2–L3 spinal puncture for spinal anesthesia using a 26G spinal needle. Doses of bupivacaine were 12 mg, 2.4 mL of bupivacaine 0.5%. The mean spread of analgesia was to T3, which was reached in 10–15 min. To prevent hypotension, we provided 500 to 1000 mL of crystalloid preload in SA. In order to prevent aortocaval compression during GA, the patient is positioned with a 15° left lateral tilt on the operating table, and 100% oxygen is administered for two minutes. The injection of opioids prior to the commencement of general anesthesia may be a problem in a cesarean section. 

Paracetamol i.v. was administered to control the postoperative pain and avoid the administration of opioids in the postoperative period. To avoid the use of opioids in the postoperative phase, 1 bag of 100 mL infusion (10 mg paracetamol/mL infusion solution) was given intravenously every 6–8 h for 24 h, as needed. Patients who required opioid administration were excluded from the research.

### 2.2. Sample Collection

Samples were taken from all subjects, pre-operatively (pre-OP), at 6 h postoperatively (6 h post-OP), and 12 h postoperatively (12 h post-OP). We collected blood samples using a venous method into tubes without any additives (Vacutest Kima, Arzegrande, Padova, Italy). We held the tubes upright for 30 min at room temperature to allow a clot to form, and then centrifuged them (Hermle AG, Gosheim, Baden-Württemberg, Germany) for 10 min at 3000× *g*. Following the removal of the clot, sera were collected in a number of cryotubes and kept at temperatures less than 20 °C for evaluation.

We left the cryotubes at room temperature while processing the samples, not allowing the leftover samples to refreeze.

### 2.3. Immunological Investigations

The Immunology Laboratory of the UMPh of Craiova conducted immunological investigations. The technique employed was Enzyme-Linked Immunosorbent Assay (ELISA), quantitative sandwich variant, following the manufacturer instructions.

We used dedicated kit tests for each of the mediators: TNF-α, IL-8 (Catalog No: E-EL-H0109, E-EL-H6008; Sensitivity: 4.69 pg/mL; Detection Range: 7.81–500 pg/mL); IL-6, IL-10 (Catalog No:E-EL-H6156, E-EL-H6154; Sensitivity: 0.94 pg/mL; Detection Range: 1.56–100 pg/mL); IL-4, EPI (Catalog No:E-EL-H0101, EL-H0045; Sensitivity: 18.75 pg/mL; Detection Range: 31.25–2000 pg/mL), NE (Catalog No:E-EL-H0047; Sensitivity: 0.19 ng/mL; Detection Range: 0.31–20 ng/mL), Elabscience (Houston, TX, USA).

Dilutions and working processes were carried out in accordance with manufacturer instructions and recommended methods. 

The ELISA method was used with a standard optical analyzer with a 450 nm wavelength (Asys Expert Plus UV G020 150 Microplate Reader, ASYS Hitech GmbH, Eugendorf, Austria).

### 2.4. Ethical Issue

The ethical aspects of the scientific research were respected based on the patients’ informed agreement. The Ethics Committee of the UMPh of Craiova, No. 135/17 September 2021, approved the study. 

### 2.5. Statistical Analysis

Microsoft Excel was used to manage and process data collected from medical documents for patients. GraphPad Prism 5 version (San Diego, CA, USA) was used to analyze the data statistically.

The D’Agostino and Pearson Omnibus Normality Test has been used to test data normality. Biomarkers that showed a normal distribution, such as TNF-α, IL-8, IL-4, NE and EPI, were expressed as mean values accompanied by the standard deviation (SD). Instead, IL-6 and IL-10 showed an abnormal distribution, being thus expressed by the median accompanied by the interquartile range (IR). Categorical data were reported as a percentage.

The one-way ANOVA estimated the difference between the groups used to analyze the differences between the groups for parametric variables. In contrast, the Kruskal–Wallis’s test was used for non-parametric variables. The statistical threshold was 5%, and for *p* ≤ 0.05 values, the results were considered significant.

Significant correlations between catecholamine levels (NE and EPI) and inflammatory status biomarkers (TNF-α, IL-6, IL-8, IL-4, IL-10) were evaluated using Spearman’s coefficients (−1 < rho < 1). The correlation heatmap matrix was used to visually represent the results, with colours ranging from brilliant red for strong negative correlations to bright green for strong positive correlations.

## 3. Results

### 3.1. Clinical Characteristics of the Investigated Patients

We could not find statistically significant differences between the two groups of patients since the demographic data was identical, regardless of age or urban/rural areas (Table 1).

Using the ASA Physical Status categorization level to assess the medical comorbidities of the patients before to anesthesia, we discovered that ASA I predominated in both analyzed groups (GA group—10 cases, 66.67%; SA group—9 cases, 60%).

### 3.2. Group Comparisons of the Cytokines, and Catecholamines

Using the one-way ANOVA test, we obtained that both patient groups (SA and GA) had significantly higher levels of three of the inflammatory mediators tested (TNF-α, IL-6, and IL-4) at 6 h and 12 h post-OP compared to pre-OP levels. Also, the Kruskal–Wallis’s test revealed significantly higher IL-8 and IL-10 levels at 6 h and 12 h post-OP. Comparing the serum concentrations of cytokines between the two groups, we found that at the time of pre-OP only for IL-6 and IL-4 were there significantly increased levels in the GA versus SA group. On the other hand, when we analyzed the other two collection times, at 6 h and 12 h post-OP, for all the investigated cytokines, we obtained significantly higher serum concentrations in the GA versus SA group (Table 2).

In the case of catecholamines, we obtained evidence that both groups of patients (SA and GA) expressed a significant statistically significant increase in levels only for NE, at 6 h and 12 h post-OP compared to pre-OP levels. The EPI showed significantly increased levels only at 12 h post-OP versus pre-OP time. Comparing the serum concentrations of catecholamines between the two groups, we found that both at the time of pre-OP and at the other two collection times, at 6 h and 12 h post-OP, for both investigated catecholamines, we obtained significantly higher serum concentrations in the GA versus SA group (Table 3).

### 3.3. Catecholamines Serum Levels Associated Positively with Cytokines

In the GA group, at 6 h post-OP (Figure 2A), the Spearman’s test showed a statistically significant correlation between the NE and pro-inflammatory cytokines levels, IL-6 and IL-8 (moderate positive correlations, rho = 0.493, *p* < 0.0001, and rho = 0.369, *p* = 0.004, respectively). Additionally, the EPI values exhibited a weak positive correlation with TNF-α (rho = 0.268, *p* = 0.053), reaching the limit of significance.

One important finding from our study was that both catecholamines had weak–negative relationships with IL-4, a cytokine that helps fight inflammation (rho = −0.431, *p* < 0.0001, and rho = −0.357, *p* = 0.004, respectively). 

At 12 h post-OP (Figure 2B), the NE values correlated moderately and significantly with TNF-α (rho = 0.464, *p* = 0.007) and IL-6 (rho = 0.396, *p* = 0.048), but also with the anti-inflammatory cytokine, IL-10 (rho = 0.308, *p* = 0.026). Compared to 6 h post-OP time, at 12 h post-OP time, we observed that EPI values correlated moderately with both anti-inflammatory cytokines, IL-4 and IL-10 (rho = 0.336, *p* = 0.035, and rho = 0.295, *p* = 0.014, respectively).

In the SA group, at 6 h post-OP (Figure 3A), the Spearman’s test indicated a statistically significant correlation only between NE and IL-8 (moderate positive correlation, rho = 0.464, *p* = 0.034) and IL-4 (moderate negative correlation, rho = −0.624, *p* = 0.027).

Also, for 12 h post-OP (Figure 3B), the statistical test showed the same finding, as among the two investigated catecholamines, only NE presented weakly but positively statistically significant correlations with pro-inflammatory cytokines, TNF-α (rho = 0.198, *p* = 0.027), IL-6 (rho = 0.143, *p* = 0.037), and IL-8 (rho = 0.308, *p* = 0.020), and moderate correlation with IL-4 (negative correlation, rho = −0.502, *p* = 0.046).

## 4. Discussion

Anesthetic drugs, along with other stressors, have a direct or indirect effect on immune system function. In this sense, they exhibit strong immunosuppressive properties, both anti-inflammatory and inflammatory [16]. There is a documented interaction between the immune system and opioids, characterized by both activating and suppressive actions, but this is not yet fully understood [29,30,31]. 

We did not administer opioids in the postoperative phase in order to accurately reflect the immunological condition. Instead, we used intravenous paracetamol to treat postoperative pain.

In a previous study [15] which focused on the immunological effect of anesthetic drugs used in minimally invasive surgery, we showed that in the case of GA with Propofol, no significant statistical significance occurred in the plasma values of TNF-α and IL-6 at 2 h postoperatively.

Cytokines are small proteins known as glycoproteins or polypeptides that are released by cells and regulate the growth, maturation, and response of certain cell populations through a variety of receptors [32,33]. They contribute to acute and/or chronic inflammatory processes via a variety of interactions, resulting in an exceedingly complex immunological network.

Pro- and anti-inflammatory cytokines play an important immunomodulatory role in reducing the likelihood of injury or an excess of inflammatory reactions under normal conditions. In pathological conditions, their imbalance can result in a systemic inflammatory response that is either too high or too low. In contrast, a healthy and dynamic balance of pro- and anti-inflammatory cytokines reduces organ dysfunction, infection, and immunity while also contributing to the body’s healing process [34,35].

As a result, our goal with each medical procedure is to identify the best approach to a minimally invasive procedure that allows us to control the release of these cytokines and catecholamines, thereby preventing their impact on important internal systems. Studies have demonstrated that certain medications like opioids and propofol can decrease the release of vasoactive hormone and inflammatory cytokines [36,37].

The current investigation discovered statistical significance only for IL-6 and IL-4 in the GA versus SA groups at the time of pre-OP. At 6 h and 12 h post-OP, serum concentrations of TNF-α, IL-6, IL-8, IL-4, and IL-10 were significantly greater in the GA group compared to the SA group. However, cytokine levels increased in both study groups. We proposed that factors other than surgical technique influenced cytokine modulation, given the surgical and hospitalization stress, increased blood loss after cesarean section and tissue injury.

Vosoughian et al. [26] also found an increase in cytokine levels in the GA group compared to the SA group in patients with pre-eclampsia undergoing cesarean delivery. Based on these findings, we recommend utilizing SA for cesarean sections to reduce the elevated cytokine production following surgery. To support this idea, we provide the findings from the correlation analysis. Compared to the SA group, we observed that both catecholamines correlated significantly better with pro-inflammatory cytokines at 6 h post-OP and with both types of cytokines at 12 h post-OP. It is interesting to note that the majority of the correlations were positive and moderate in intensity, as opposed to the weak correlations identified in the SA group for both harvest times, 6 h and 12 h post-OP. This is because an excessive inflammatory response can produce postoperative problems such as systemic inflammatory response syndrome, hemodynamic abnormalities, and delayed healing of the surgical incision. The SA affects the motor, sensory, and autonomic nervous systems by blocking the autonomic nervous system. This activity minimizes inflammatory stress and can totally block all relevant neurogenic impulses, but it also prevents neuroendocrine activation during surgery [38]. 

Regarding opioids provided intraoperatively, all of our GA patients received Fentanyl during the anesthetic induction. However, the elimination half-life of 475 min leads us to believe that Fentanyl administration had no effect on the regulation of cytokine production during collection times. Other researchers [27,39,40] discovered that in GA, the level of inflammatory cytokines increases and is regulated by surgical stress. It appears that the duration of the surgery dictated the inflammatory cytokine response rather than Propofol, which did not appear to alter their reaction, despite Propofol’s anti-inflammatory activity. As a result, in our analysis, we excluded cases with a cesarean section extending longer than 2 h.

Unlike our findings, Hassanshahi et al. [41] and Andreis et al. [42] showed higher levels of IL-6 and TNF-α in the SA group compared to GA [41], or equivalent levels of IL-6 in both anesthetic procedures [42]. However, it should be emphasized that not all research used the same immunological characteristics, sample collection intervals, types of patients undergoing cesarean sections (emergency or elective), or pharmacological pain treatment. That is why, to improve the accuracy of the data, our study focused only on elective cesarean sections to avoid an increase in pain-related cytokines during labor prior to surgical intervention.

Stress hormones known as catecholamines are released into the body in response to various stimuli and play an important part in the autonomic nervous system’s ability to maintain homeostasis [43]. 

Catecholamines affect all bodily tissues and have a considerable impact on neurological, endocrine, metabolic, and cardiac functions. These effects also have an impact on the intestinal barrier, which in turn affects the immune response [44].

Another connection that we investigated to gain a better understanding of how the immune system works was the relationship between this system and the sympathetic nervous system in terms of surgical and anesthetic stress. Surgical stress causes the release of catecholamines (NE and EPI), adrenocorticotropic hormone, and cortisol in the autonomic nervous system. Researchers have reported that surgical stress decreases immunological processes, promotes cell signaling, and reduces the production of inflammatory cytokines [45].

In general anesthesia, anesthetic drugs can decrease the surgical stress response by inhibiting the hypothalamic-pituitary-adrenal (HPA) axis and decreasing the production of EPI and NE, thus serving as an immunoprotectant [46,47]. 

Our investigation found that both the GA and SA groups had higher NE levels at 6 h and 12 h post-OP compared to pre-OP time. Increased responses to NE stress have been associated with an increase in inflammatory responses [48].

In contrast, EPI increased only at 12 h post-OP compared to pre-OP. When we compared serum NE and EPI concentrations at pre-OP, 6 h post-OP, and 12 h post-OP, we found statistical significance in the GA versus SA groups. We specifically excluded cases requiring EPI or NE to treat spinal anesthesia-induced hypotension during cesarean delivery from the research. We did this precisely to provide results that were as unaffected as possible.

According to Ferland et al. [13], serum NE values remain elevated 24 h after surgery or 72 h following GA [49], although plasma EPI values normalize rapidly. In accordance with these published results, we provide our study’s data, which revealed an increase in the plasma value of NE up to 12 h during the last examination, whereas EPI showed only a modest increase in the plasma level at 12 h. However, we have only partially established the practical significance of the postoperative catecholamine response. Researchers have discovered that an increase in cortisol might impair the activity of NK cells, thereby slowing down healing and recovery. Despite our efforts to obtain study groups with minor interferences (healthy pregnancies, no postoperative opioid administration, cases with elective cesarean sections only, exclusion of cases with spinal anesthesia-induced hypotension) and homogeneous groups, it appears that there are a number of biological and genetic factors that can cause an increase in the nociceptive threshold [50,51,52].

This study’s strength is that we used a homogeneous sample of patients. We only looked at patients who had elective cesarean sections because we believed that, in the case of emergency cesarean patients, the pain induced by labor progression could impact cytokine production. Another interesting component of this study was the inclusion of healthy patients who had no interference from other disorders that could have altered the immunomodulation. Due to funding constraints, a limitation of this study was that the serum levels of cytokines were only evaluated in 30 patients (15 GA and 15 SA), and only 5 cytokines were studied despite the fact that the number of cytokines was higher. Another limitation of the study would be the fact that the selection of patients would have determined a certain bias between the groups, which may affect the statistical significance of the results obtained in our study.

## 5. Conclusions

The results of this study reaffirm the role of cytokines and catecholamines in postoperative immunomodulation in both GA and SA. However, SA appears to have fewer impacts on the immunological response, causing a milder inflammatory response. As a result, SA for cesarean section may be a better alternative as an anesthetic procedure that reduces perioperative stress and the administration of opioids, influencing the low production of cytokines and implicit immunomodulation. 

## Figures and Tables

**Figure 1 diagnostics-14-00880-f001:**
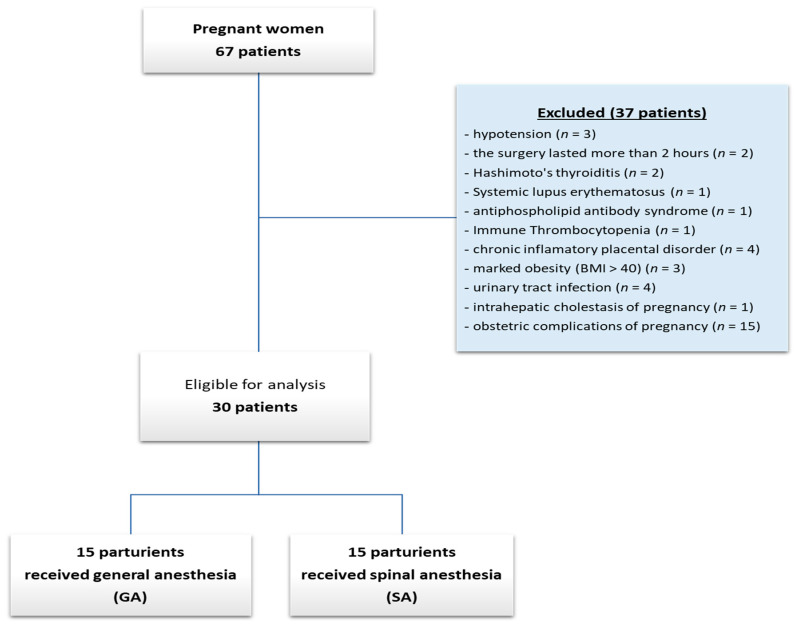
Flow diagram with patients included.

**Figure 2 diagnostics-14-00880-f002:**
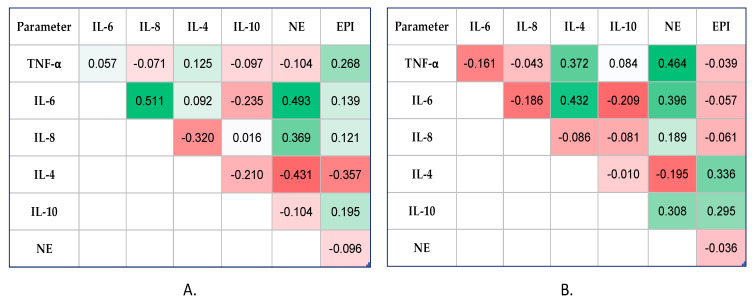
Correlation heatmap matrix between measured catecholamines and cytokines (colours ranging from brilliant red for strong negative correlations to bright green for strong positive correlations) in GA group: (**A**) At 6 h post-OP time, and (**B**) at 12 h post-OP time. TNF-α: tumor necrosis factor-alpha; IL: interleukin; NE: norepinephrine; EPI: epinephrine.

**Figure 3 diagnostics-14-00880-f003:**
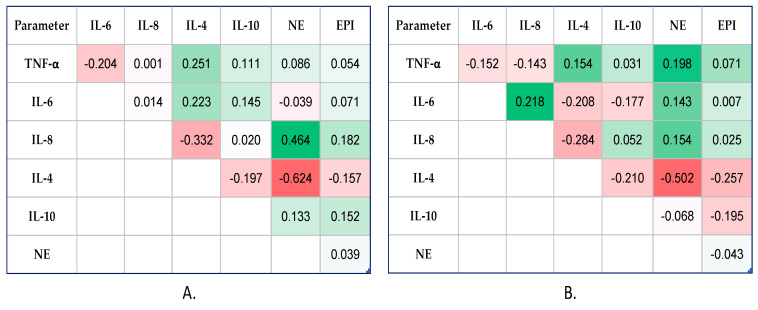
Correlation heatmap matrix between measured catecholamines and cytokines (colours ranging from brilliant red for strong negative correlations to bright green for strong positive correlations) in SA group: (**A**) At 6 h post-OP time, and (**B**) at 12 h post-OP time. TNF-α: tumor necrosis factor-alpha; IL: interleukin; NE: norepinephrine; EPI: epinephrine.

**Table 1 diagnostics-14-00880-t001:** Patient demographics and clinical features.

Characteristics	GA Group (*n* = 15)	SA Group (*n* = 15)
Age (yrs) (mean ± SD)	32.3 ± 4.82	31.2 ± 5.41
BMI (Body Mass Index) (kg/m^2^) (mean ± SD)	23.14 ± 3.45	21.68 ± 4.72
Urban/rural areas	8/7	7/8
ASA (American Society of Anesthesiologists) grade I	10 (66.67%)	9 (60%)
ASA II	5 (33.33%)	6 (40%)
Gravidity	2 ± 0.48	2 ± 0.83
Parity	2 ± 0.45	2 ± 0.48
Gestational age at delivery (weeks)	38.3 ± 0.62	38.5 ± 0.60
Birth weight (g)	3355 ± 165.19	3157 ± 207.10
Apgar score at 1 min	8 ± 0.63	9 ± 0.83
Apgar score at 5 min	9 ± 0.63	10 ± 0.45
ICU (Intensive Care Unit) admission (*n*/%)	3 (20%)	2 (13.3%)
Preoperative hemoglobin (g/100 mL)	12.2 ± 0.76	12.3 ± 0.45
Postoperative hemoglobin (g/100 mL)	10.8 ± 0.78	11.2 ± 0.44
Preoperative hematocrit (%)	34.2 ± 1.45	34.1 ± 1.20
Postoperative hematocrit (%)	30.1 ± 1.20	32.4 ± 0.73
Surgical time (minutes)	71.5 ± 10.70	52.7 ± 6.99
Time to first mobilization *(n*/%)		
6 h	2 (13.33%)	4 (26.66%)
6–12 h	8 (53.33%)	11 (73.33%)
12–24 h	5 (33.33%)	0
Postoperator complication		
Nausea/vomiting	2 (13.33%)	1 (6.66%)
Headache	0	3 (20%)

**Table 2 diagnostics-14-00880-t002:** Comparison of serum cytokine concentrations pre-operatively, at 6 h and 12-h postoperatively, in the two groups.

Cytokines (pg/mL)	Time	GA Group (*n* = 15)	SA Group (*n* = 15)	*p*-Value
TNF-α (Mean ± SD)	pre-OP	7.11 ± 1.52	6.68 ± 2.05	ns
	6 h post-OP	12.00 ± 2.18	10.30 ± 1.08	*
	12 h post-OP	17.50 ± 5.46	13.70 ± 2.21	*
IL-6 [Median (IR)]	pre-OP	5.68 (4.98–6.21)	3.33 (3.04–3.66)	**
	6 h post-OP	11.00 (9.21–13.00)	6.89 (5.96–7.09)	**
	12 h post-OP	15.40 (14.60–17.20)	9.66 (8.74–10.20)	**
IL-8 (Mean ± SD)	pre-OP	45.80 ± 14.15	43.60 ± 13.80	ns
	6 h post-OP	108.00 ± 33.00	62.70 ± 17.00	**
	12 h post-OP	162.00 ± 62.70	92.00 ± 27.70	**
IL-4 (Mean ± SD)	pre-OP	84.90 ± 11.80	73.80 ± 10.30	**
	6 h post-OP	115.00 ± 16.00	99.70 ± 13.90	**
	12 h post-OP	143.00 ± 20.00	125.00 ± 17.40	**
IL-10 [Median (IR)]	pre-OP	4.14 (3.68–5.00)	3.91 (3.76–4.54)	ns
	6 h post-OP	5.62 (5.41–6.53)	4.49 (4.33–5.22)	**
	12 h post-OP	8.22 (7.91–9.55)	5.67 (5.46–6.59)	**

pre-OP: pre-operatively; GA: general anesthesia; 6 h post-OP: at 6 h postoperatively; 12 h post-OP: at 12 h postoperatively; IL-4: interleukin-4; IL-6: interleukin-6; IL-8: interleukin-8; IL-10: interleukin-10; SA: spinal anesthesia; TNF-α: tumor necrosis factor-alpha; SD: Standard deviation; IR: interquartile range; Kruskal–Wallis/one-way ANOVA tests were used to determine statistical significance between groups. *: *p* < 0.05; **: *p* < 0.0001; ns: not statistically significant.

**Table 3 diagnostics-14-00880-t003:** Comparison of serum catecholamine concentrations pre-operatively, at 6-, and 12-h postoperatively, in the two groups.

Catecholamines (Mean ± SD)	Time	GA Group (*n* = 15)	SA Group (*n* = 15)	*p*-Value
NE (ng/mL)	pre-OP	4.85 ± 0.35	3.13 ± 0.23	*p* < 0.0001
	6 h post-OP	6.40 ± 0.98	4.16 ± 0.42	
	12 h post-OP	8.76 ± 1.13	5.33 ± 0.51	
EPI (pg/mL)	pre-OP	498.00 ± 151.00	399.00 ± 121.00	
	6 h post-OP	598.00 ± 181.00	478.00 ± 145.00	
	12 h post-OP	747.00 ± 226.00	598.00 ± 181.00	

EPI: epinephrine; GA: general anesthesia; pre-OP: pre-operatively; 6 h post-OP: at 6 h postoperatively; 12 h post-OP: at 12 h postoperatively; NE: Norepinephrine; SA: spinal anesthesia; SD: standard deviation.

## Data Availability

The data used to support the findings of this study are available from the corresponding author upon reasonable request.

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
