# Peer review of "Immune Responses of Healthy Pregnant Women following an Elective Cesarean Section: Effects of Anesthetic Procedures"

_diagnostics, 2024, doi:10.3390/diagnostics14090880_

Round 1

Reviewer 1 Report

Comments and Suggestions for Authors

MAJOR ISSUES

It is not clear how women were selected for either GA or SA.

It is remarkable that pre-OP catecholamines and cytokines are consistently lower (although not always statistically significant) in the SA group. Is this a result of selection bias due to selection criteria to decide on either GA or SA, or a difference in premedication?

ABSTRACT

Line 35 BAI is an unknown abbreviation, possibly pre-OP makes it easier to remember its meaning

You summarize all results, which is rather difficult to read without further interpretation. Possibly you can shorten this and restrict text to the most relevant findings. You should end by explaining the importance of your findings (like in the conclusion).

INTRO

Line 79: I assume that tissue damage by the surgical procedure, and the resulting inflammatory response, is also important in this respect.

METHODS

Line 130: You may change inclusion criteria description to: “The study included healthy (ASA Physical Status I and II ) women with a singleton pregnancy, who were scheduled before the onset of labor for an elective cesarean section for maternal and perinatal risk.” Then exclusion from analysis was when spinal anesthesia caused hypotension, or  difficult intubation occurred, or operative procedure lasted more than 2 hrs (this occurs very rarely in elective CS). Thus you should show how many women were included pre-operative and how many were excluded afterwards because of complications.

Why were women not randomized? If not, how did you choose between GA and SA

Any pre-medication?

You specify standard paracetamol administration iv. Was this given immediately after operation, and continued orally thereafter? Were opioids given additionally if needed? In the discussion you write that no opioids were given. This should also specified in the methods. What did you prescribe if women complained of pain within the 12 hours postop period?

Any difference in oral intake or mobilization between GA and SA.

Any postop complications?

Did you perform a power analysis.

RESULTS

You should provide more obstetric information, like parity, repeat CS, indication for CS, gestational age, birth weight, preeclampsia. Some data of the CS procedure should be presented, like duration of operation and blood loss.

Table 2 and 3 show the same data. I prefer table 3 and would omit table 2. You could specify significant differences from pre-OP by a symbol behind the measurement value. Same for table 4 and 5

CONCLUSION

Line 395: “if opioids are not used” – you did not compare use of opioids yes or no, so this statement is not supported by your data.

The relevance of the last sentence is not clear to me.

Comments on the Quality of English Language

See above

Author Response

Dear Reviewer,

Thank you very much for taking the time to analyze our manuscript, as well as for your kind appreciation and valuable suggestions.

All the recommended changes were performed in the body of our manuscript, with the Track Changes function activated.

Comments and Suggestions for Authors

MAJOR ISSUES

It is not clear how women were selected for either GA or SA.

The patients were selected according to some criteria that we added in 2.1. Study Design and Patient Selection

It is remarkable that pre-OP catecholamines and cytokines are consistently lower (although not always statistically significant) in the SA group. Is this a result of selection bias due to selection criteria to decide on either GA or SA, or a difference in premedication?

The patients were selected according to the selection criteria, and being healthy pregnant women, they benefited from the same premedication, described and presented in 2.1. Study Design and Patient Selection

ABSTRACT

Line 35 BAI is an unknown abbreviation, possibly pre-OP makes it easier to remember its meaning

Revised :  BAI, replaced it with pre-OP according to the relevant recommendation

You summarize all the results, which is rather difficult to read without further interpretation. Possibly you can shorten this and restrict text to the most relevant findings. You should end by explaining the importance of your findings (like in the conclusion).

Revised Abstract according to the recommendations

INTRO

Line 79: I assume that tissue damage by the surgical procedure, and the resulting inflammatory response, is also important in this respect.

Revised : added to the Introduction, according to the suggestions

METHODS

Line 130: You may change inclusion criteria description to: “The study included healthy (ASA Physical Status I and II ) women with a singleton pregnancy, who were scheduled before the onset of labor for an elective cesarean section for maternal and perinatal risk.” Then exclusion from analysis was when spinal anesthesia caused hypotension, or  difficult intubation occurred, or operative procedure lasted more than 2 hrs (this occurs very rarely in elective CS). Thus you should show how many women were included pre-operative and how many were excluded afterwards because of complications.

Revised : included the recommendations in 2.1.Study Design and Patient Selection

Why were women not randomized? If not, how did you choose between GA and SA

Any pre-medication?

Yes. As the prevention of hypotension, we used the administration of a 500 to 1000 mL crystalloid preload in SA, and in GA, to avoid aortocaval compression, the patient is positioned with left lateral inclination, with the operating table with a left lateral tilt of 15°. The administration of opioids before induction of general anesthesia can be considered as a problem in cesarean section. I added to 2.1. Study Design and Patient Selection

You specify standard paracetamol administration iv. Was this given immediately after operation, and continued orally thereafter? Were opioids given additionally if needed? In the discussion you write that no opioids were given. This should also specified in the methods. What did you prescribe if women complained of pain within the 12 hours postop period?

Added and described in 2.1. Study Design and Patient Selection the way of administering the Paracetamol solution. Patients who also needed opioids were subsequently excluded from the study.

Any difference in oral intake or mobilization between GA and SA.

We did not administer Paracetamol oral intake, only i.v.

Mobilization was earlier in SA compared to GA.

Any postop complications?

Yes: Headache - Headache in cases with SA.

There were no massive hemorrhages or other intraoperative interventions.

Did you perform a power analysis.

RESULTS

You should provide more obstetric information, like parity, repeat CS, indication for CS, gestational age, birth weight, preeclampsia. Some data of the CS procedure should be presented, like duration of operation and blood loss.

Revised : we presented the recommended data by completing Table 1. Patient demographics and clinical features.

Table 2 and 3 show the same data. I prefer table 3 and would omit table 2. You could specify significant differences from pre-OP by a symbol behind the measurement value. Same for table 4 and 5.

Revised : according to the suggestions

CONCLUSION

Line 395: “if opioids are not used” – you did not compare use of opioids yes or no, so this statement is not supported by your data.

Revised : modified the phrase

The relevance of the last sentence is not clear to me.

Revised : modified the phrase

Reviewer 2 Report

Comments and Suggestions for Authors

Dear authors!

The topic of the article is very important for understanding pain and opiates in the regulation of the immune response and

Minor errors in the text should be corrected, and the consistency of the drawings and signatures should be checked.

There are quite a lot of references in the bibliography that are older than 15 years; it is advisable to replace them with key ones.

Comments on the Quality of English Language

It is advisable to rework the article, taking into account that English is not the native language of the authors. Correct minor errors, e.g. "The rol of cytokines in the acute inflammatory response initiated by surgical trauma"... Line 95

Author Response

Dear Reviewer,

Thank you very much for taking the time to analyze our manuscript, as well as for your kind appreciation and valuable suggestions.

All the recommended changes were performed in the body of our manuscript, with the Track Changes function activated.

Comments and Suggestions for Authors

Dear authors!

The topic of the article is very important for understanding pain and opiates in the regulation of the immune response and

Minor errors in the text should be corrected, and the consistency of the drawings and signatures should be checked.

There are quite a lot of references in the bibliography that are older than 15 years; it is advisable to replace them with key ones.

Revised : we replaced the bibliography according to the recommendations

Comments on the Quality of English Language

It is advisable to rework the article, taking into account that English is not the native language of the authors. Correct minor errors, e.g. "The rol of cytokines in the acute inflammatory response initiated by surgical trauma"... Line 95

Revised : according to the suggestions.

Round 2

Reviewer 1 Report

Comments and Suggestions for Authors

My questions regarding how women were selected for either SA or GA was not answered, and selection of women for either SA or  GA remains unclear. The observation that pre-OP catecholamines and cytokines are consistently lower (although not always statistically significant) in the SA group than in the GA group suggests that this selection caused a certain bias between the groups, which might invalidate the results. This should at least be mentioned in the discussion.

The last sentence of the conclusion is an opinion. The statements are not supported by the manuscript. This sentence should therefore be removed.

Comments on the Quality of English Language

No comment

Author Response

Dear Reviewer,

Thank you very much for taking the time to analyze our manuscript, as well as for your kind appreciation and valuable suggestions.

All the typing recommended changes were performed in the body of our manuscript, with the Track Changes function activated.

Comments and Suggestions for Authors

My questions regarding how women were selected for either SA or GA was not answered, and selection of women for either SA or GA remains unclear.

  • The patients were selected according to some criteria that we added in 2.1. Study Design and Patient Selection:

The indication criteria for the type of anesthesia used were: for GA, regional contraindications including spinal abnormalities (e.g. spina bifida, scoliosis), inadequate or failed regional attempts, a history of hypersensitivity to local anaesthetic, maternal refusal of regional techniques; for SA, which became the preferred anesthetic technique by many anesthesiologists in elective conditions, the indications were, history of hypersensitivity to the study drugs used in GA, patients at risk of difficult intubation, maternal refusal of GA, the mother's desire to remain awake during the birth, for an immediate interaction with the newborn, and all other cases that did not fall into the GA category.

The observation that pre-OP atecholamines and cytokines are consistently lower (although not always statistically significant) in the SA group than in the GA group suggests that this selection caused a certain bias between the groups, which might invalidate the results. This should at least be mentioned in the discussion.

  • Revised - added to the Discussion, according to the suggestions, the sentence:

Another limitation of the study would be the fact that the selection of patients would have determined a certain bias between the groups, which may affect the statistical significance of the results obtained in our study.

The last sentence of the conclusion is an opinion. The statements are not supported by the manuscript. This sentence should therefore be removed.

  • Revised according to the suggestions : removed
